# "I don't know if it makes a difference to safety?" perception vs actuality: A mixed-methods study on older adults' experiences of home stair falls revealed during COVID-19 lockdown

Emily Wharton[1,2]*, Thomas O'Brien[1], Richard J. Foster[1], Clarissa Giebel[2,3], Justine Shenton[4], Asan Akpan[5], Avril Mills[2], Mike Roys[6], Constantinos Maganaris[1]

1 School of Sport and Exercise Science, Faculty of Science, Liverpool John Moore's University, Liverpool, United Kingdom, 2 NIHR ARC NWC, Liverpool, United Kingdom, 3 Department of Primary Care & Mental Health, University of Liverpool, Liverpool, United Kingdom, 4 Sefton Older People's Forum, Sefton, United Kingdom, 5 Liverpool University Hospital NHS F.T., Liverpool, United Kingdom, 6 Rise and Going Consultancy, Watford, United Kingdom

* e.h.wharton@2021.ljmu.ac.uk

## Abstract

In the United Kingdom (UK), stair falls in older adults' homes cause up to 575 deaths and 350,000 injuries annually, costing the NHS £435 million. The stair falls may be related to hazards such as poorly designed/absent handrails, too steep/narrow stairs, poor step surface (e.g., loose carpets), and poor lighting. Our study aimed to understand older adults' experiences of independent living and home stair falls during the first COVID-19 lockdown, and shed light on older adults' physical staircase dimensions that influence stair fall risks in relation to UK government guidelines. A mixed-methods approach was employed, conducting semi-structured interviews alongside quantitative home stair assessments with 22 participants aged ≥ 60 years. The stair assessments captured the physical dimensions (i.e., measurements of pitch, rise and goings) of their home stairs, and if they perceived their stairs safe to negotiate. We identified four overarching themes common across older people living independently: effects of lockdown on daily living during the COVID-19 pandemic; stair-related accidents and perceived causes; fall preventative measures and safety awareness; and attitudes towards ageing and care services. Although all of the participants perceived their stairs to be safe, nearly half of participants' staircases (40%) did not meeting the UK government guidelines for pitch, rise and going. While the COVID-19 lockdown provided a unique context for exploring fall risk and stair safety, our findings highlight broader, ongoing issues. Despite emotional attachment to their homes, many lacked staircases that meet current UK government guidelines, highlighting a need for targeted interventions to mitigate environmental hazards. Additionally, financial constraints and education further complicate efforts to enhance home safety. Discrepancies between perceived and objective safety assessments highlight

**Data availability statement:** Due to ethical and privacy considerations outlined by the LJMU UREC Ethics Committee, the authors are unable to make the data publicly available. However, the data are accessible upon reasonable request and subject to approval by the Liverpool John Moores University Research Ethics Committee. Requests for access can be directed to the Liverpool John Moores University Research Ethics Committee via email (researchethics@ljmu.ac.uk) for researchers who meet the criteria for access to confidential data.

**Funding:** Initials of the authors who received each award: EW (NIHR200182) National Institute for Health and Care Research Applied Research Collaboration North West Coast https://arc-nwc.nihr.ac.uk/. This independent research is funded by the National Institute for Health and Care Research Applied Research Collaboration North West Coast (ARC NWC) as part of a PhD studentship. The views expressed in this publication are those of the author(s) and not necessarily those of the National Institute for Health and Care Research or the Department of Health and Social Care. The funders had no role in study design, data collection and analysis, decision to publish and preparation of the manuscript.

**Competing interests:** The authors have declared that no competing interests exist.

the need for comprehensive care approaches and evidence-based home design guidelines, allowing for collaborative support for ageing in place. Bridging this gap is essential for reducing home stair falls.

## Introduction

Falls on domestic staircases in the United Kingdom (UK) represent a significant public health concern: in the year 2000 it was estimated that stair falls caused up to 575 deaths and 350,000 injuries annually [1]. While this may not immediately seem like a crisis in a nation of 67 million and over 11 million individuals over 65 (18.6% of the total population), it becomes an important issue when considering that falls are the second leading cause of accidental death among older adults. Stair falls can lead to a loss of independence, greater isolation and depression, reduced mobility, and increased morbidity and mortality [2]. Stair falls also have social impact due to the increased healthcare costs and demands on healthcare professionals. Unaddressed fall hazards in the home environment are estimated to cost the NHS in England £435 million [2]. Such unaddressed hazards may include poorly designed or absent hand-rails, stairs that are too steep and narrow, step surfaces or covering in poor condition, objects left on steps, and poor lighting [1]. Therefore, housing conditions, and specifically the staircase environment, pose a significant risk for stair falls [3]. Unfortunately, the scale of the problem is set to increase as the UK's older population continues to grow, alongside the UK possessing one of the oldest housing stocks in Europe [4,5]. Therefore, understanding the need for adaptations and improvements in the individual's staircase and surrounding environment is vital for the changing needs and abilities of an older population to live safely and independently [3,6].

On March 23rd 2020, the UK Government implemented a nationwide "lockdown" in response to the COVID-19 pandemic, including a prohibition on non-essential travel and interactions with individuals outside of one's household [7]. The increased exposure to home stairs during lockdowns and restrictive measures may have amplified the risk of falls and related injuries beyond the usual levels [6]. Moreover, lockdown increased social isolation, loneliness, ageism, sedentary behaviours, delayed essential medical treatment, and increased challenges in meeting basic needs (food shopping and/or cleaning the house) [8–11]. Although studies linking stair falls and social isolation primarily use pre-COVID-19 data [12–14], these findings are particularly relevant during the pandemic, as lockdown and distancing measures likely heightened risks of loneliness, isolation, and falls [14].

Real world changes for maintaining the independence and quality of life for older adults requires an understanding of the connection between their perceptions of home safety, housing condition, and home stair falls during lockdown. Therefore, this study aimed to understand older adults' experiences of independent living and home stair falls during and after the COVID-19 pandemic. Additionally, the study aimed to shed light on their perceptions regarding the safety of their staircases and whether these perceptions align with the staircase features and design that influence stair fall risks.

## Methods

### Study design

A nested mixed-methods design was used to investigate the studies aims. The study obtained ethical approval on the 14th June 2022 from the Liverpool John Moores University (LJMU) Ethics Committee (Ref No: 22/SPS/037) prior to study commencement.

### Participants and recruitment

Twenty-two participants (9 stair-fallers, 1 near faller, and 12 non-fallers) aged ≥ 60 years agreed to a home visit to carry out stair assessments and semi-structured interviews, between September 2022 and February 2023. Consent was obtained either verbally or written at the time of the home visit (semi-structured interviews and stair assessments) and recorded in the interviewer's notes. The number of participants recruited were consistent with recommendations for semi-structured interview studies that aim to identify patterns across data (Guest, Brunce & Johnson, 2006).

Participants were recruited from a survey study previously conducted. Briefly, the online and telephone survey was conducted with 164 UK residents aged ≥ 60 years between June and October 2021, to understand the extent to which COVID-19 lockdowns increased home stair falls. The survey included collecting participants postcodes to generate an Index of Multiple Deprivation (IMD) quintiles for socio-economic status and collecting the participants falls/near-falls experiences before (June 2019–23 March 2020) and during (23 March 2020 – October 2021) the first COVID-19 UK lockdown. The participants agreed for their contact details to be stored for the purpose of contacting them about future studies. Participants who had either reported experiencing a stair fall in their home environment, reported experiencing a stair near-fall in their home environment, and reported not having experienced a home stair fall were recruited. Participants volunteering to participate in the study were contacted via email or telephone by the primary researcher (E.W.) to inform them about the study's purpose, risks, benefits, and informed consent.

### Data collection

The semi-structured interviews were conducted by E.W. either by telephone, via videoconferencing or in the person's home according to the participant's preferences. The participant then chose a suitable time for the researcher to conduct a stair assessment in their home. The majority of participants (95%) chose to conduct the semi-structured interviews and stair assessments in one home visits.

A semi-structured topic guide (see S1 Appendix) was developed in co-operation with two public advisors (A.A. and J.S.) and older adults from Women's Club forum. The semi-structured interviews explored the participant's fall experiences and how the COVID-19 pandemic had affected the participant's activities, socialising, care routines, and details about how the fall occurred. Interviews lasted between 20–60 min (35.18 ± 9.71 min) and were recorded using an encrypted digital Dictaphone.

The stair assessment was conducted after the semi-structured interviews, using a protocol (see S2 Appendix) that identified whether the participants' staircases met existing national building regulations for stair pitch, rise and going [15–17]. We captured the physical dimensions (i.e., pitch, rise and going) and visual appearance (i.e., handrails and lighting) of their staircase. To assess the visual appearance, we adapted the housing health and safety rating system (HHSRS) assessment [17] and a previously developed Stair Assessment Questionnaire (SAQ) by researcher M.R. The stair assessment required the participants to describe their staircase dimensions (i.e., steepness, narrowness, wear and tear), record their behaviour related to stair safety, such as their use of handrails, and if they perceived their stairs safe to negotiate. Photographs of the participants stairs were taken.

We also collected background data on participant age, gender, ethnic background and postcode.

## Equipment and measurements

To enhance measurement precision, a custom-made ruler (acrylic sheet) measuring 400 mm x 270 mm, marked with a scale on both long sides (Fig 1), was utilised.

The top left of Fig 1, shows the custom-made ruler (acrylic sheet) that was utilised, measuring 400 mm x 270 mm, marked with a scale on both long sides. The middle part of Fig 1 shows the rise, going, nosing, and pitch of a staircase. The right part of Fig 1 shows the process of measuring the rise and going with the custom-made ruler.

The bottom edge of the acrylic sheet was carefully aligned with the edge of the nosing, positioned close to the side of the step, to measure the length to the next nosing. An inclinometer was used to measure the angle of the acrylic sheet. This angle measurement provided the necessary data for determining the pitch of each step. Using the nosing-to-nosing length and inclination data the rise and going of each step was calculated using trigonometry.

## Data analysis

**Qualitative analysis.** The semi-structured interviews were digitally audio-recorded and transcribed verbatim and imported into NVivo (QSR International Pty Ltd). The transcripts were anonymised and checked for accuracy. The data were analysed using an inductive thematic approach (Braun & Clarke, 2021) to extract meaningful insights from the interview data. The aim of this method was to explore and identify patterns, themes, and insights that emerged directly from the data, without being constrained by pre-existing theories or predetermined coding schemes.

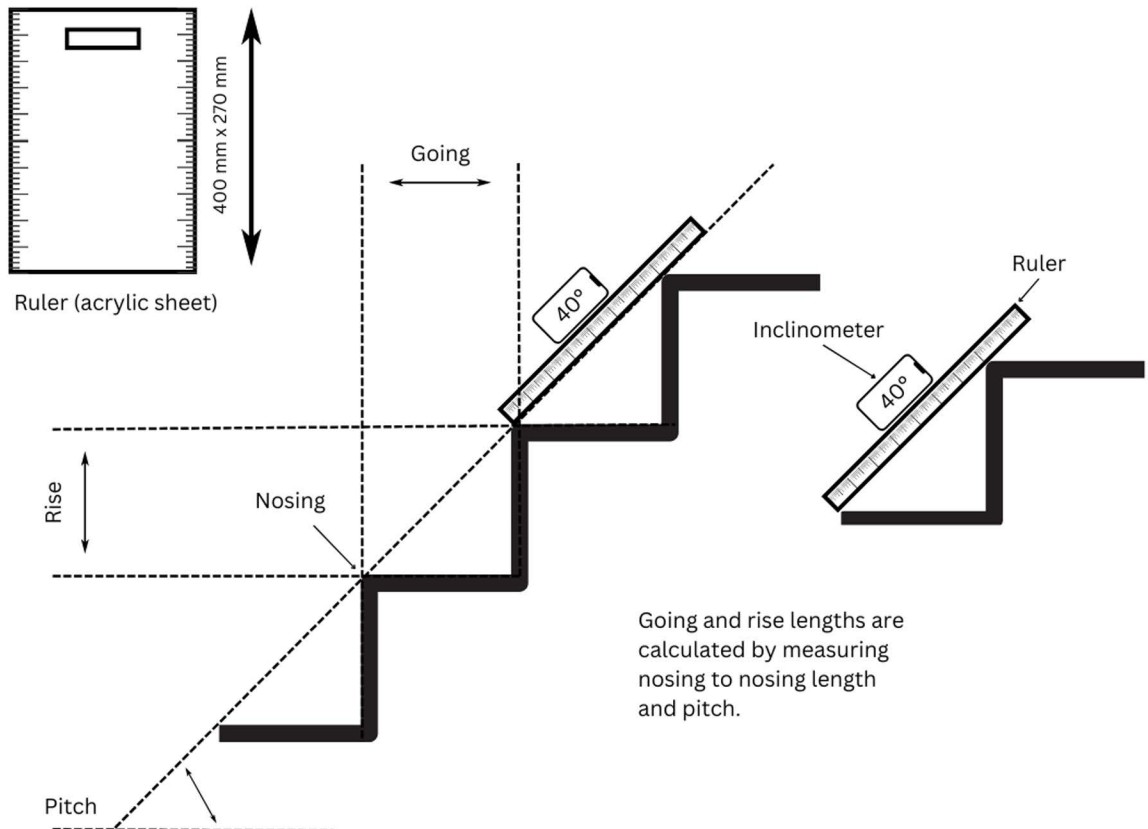

**Fig 1. The equipment used for measuring stair dimensions.**

The interview transcripts were thoroughly read to gain a comprehensive understanding of the data, immerse in the content and to identify initial impressions and ideas. After becoming familiar with the data, initial codes were generated to capture meaningful concepts, ideas, or patterns within the dataset. These codes were generated in an open-ended manner, allowing for a flexible and exploratory analysis. The initial codes were reviewed, refined, and organised into potential themes.

The data was continuously reviewed and discussed (C.M., C.G., T.O., and R.F.) until a consensus was reached on the final themes and subthemes. This iterative review process facilitated the refinement and consolidation of codes, ensuring a comprehensive and coherent representation of the interview data.

**Frequency analysis.** The stair assessment element of the study employed a frequency analysis to examine the occurrence and distribution patterns of the variables under investigation.

## Public involvement

Four public advisors (A.A., J.S., A.M., and M.R.) formed part of the research team, and assisted with conceptualising the study, designing the interview guide, interpreting the findings and with dissemination. This ensured that the study and the interpretation and implications of findings were grounded in the lived experiences of those affected by stair falls and COVID-19 lockdown. Public advisors were reimbursed for their time according to NIHR guidance.

## Results

### Demographic characteristics

Participants were all from a White ethnic background (100%), and were predominately female (68%), lived with other(s) (73%), and were retired (73%), with more participants living in less disadvantaged neighbourhoods, as measured by the IMD quintiles. A large proportion of the participants were from the North West region (82%), lived in urban major conurbation areas (77%), and owned their homes outright (91%). Additionally, the majority of participants had existing medical conditions or long-term illnesses (64%), and were in the age ranges of 70–79 years (55%). Characteristics of all participants are presented in Table 1.

### Quantitative findings

During the home stair assessment, only 20% of stair fallers stated that they were struggling to fit most of their foot on the steps, and 30% acknowledged that their stairs were steep. All stair fallers perceived their stairs as entirely safe for negotiation. However, the stair dimensions and measurements showed that 45% of staircases did not meet the safety guidelines established by the UK government, and 50% of these belonged to stair fallers.

Eight percent of stair fallers had turning stairs with quarter landings (90°/180° turn to a second flight), compared to 58% of non-fallers. Forty percent of stair fallers had patterned carpets compared to 25% of non-fallers, and 70% of stair fallers had a moveable item at the bottom of stairs compared to 30% of non-fallers. Additionally, 30% of stair fallers had items left on the steps compared to only 80% of non-fallers. Moreover, there was a difference in the use of handrails while descending stairs, with 55% of stair fallers stating that they do not used handrails compared to 45% of non-fallers (Fig 2).

### Qualitative findings

Using reflective thematic analysis, we identified four overarching themes: Effects of Lockdown on Daily Living during the COVID-19 Pandemic; Stair-Related Falls and Perceived Causes; Fall Preventative Measures and Safety Awareness; Attitudes towards Ageing and Care Services. The main findings, including quotes from participants who had experienced a fall, a near fall on the home stairs, and who had not experienced a fall on their stairs (Table 2) are presented below.

**Table 1. Demographic characteristics.**

| | Non Home Stair Fallers | Home Stair Fallers and Near-Fallers on Home Stairs |
|---|---|---|
| | (n = 12) | (n = 10) |
| *N(%)* | | |
| **Gender** | | |
| Female | 7 (58%) | 8 (80%) |
| Male | 5 (42%) | 2 (20%) |
| **Ethnicity** | | |
| White | 12 (100%) | 10 (100%) |
| People from Ethnic Minority Backgrounds | 0 (0%) | 0 (0%) |
| **Age** | | |
| 60–69 | 2 (17%) | 4 (40%) |
| 70–79 | 7 (58%) | 5 (50%) |
| 80 + | 3 (25%) | 1 (10%) |
| **IMD quintiles** | | |
| 1 (most deprived) | 3 (25%) | 1 (10%) |
| 2 | 1 (8%) | 0 (0%) |
| 3 | 3 (25%) | 4 (40%) |
| 4 | 3 (25%) | 4 (40%) |
| 5 | 3 (17%) | 1 (10%) |
| **Living situation** | | |
| Living alone | 3 (25%) | 3 (30%) |
| Living with someone | 9 (75%) | 7 (70%) |
| **Occupation** | | |
| Full-time | 1 (8%) | 0 (0%) |
| Part-time | 1 (8%) | 0 (0%) |
| Volunteer | 3 (25%) | 1 (10%) |
| Retired | 7 (58%) | 9 (90%) |
| **Home ownership** | | |
| Owned outright | 11 (92%) | 9 (90%) |
| Owned with a mortgage/loan | 1 (8%) | 1 (10%) |
| **Existing medical conditions or long-term illnesses** | 7 (58%) | 7 (70%) |
| **COVID-19 Physical Activity Levels** | | |
| 1. Significantly less physically active | 0 (0%) | 3 (30%) |
| 2. Less physically active | 3 (25%) | 3 (30%) |
| 3. No change in my physical activity level | 7 (58%) | 3 (30%) |
| 4. More physically active | 2 (17%) | 1 (10%) |
| 5. Significantly more physically active | 0 (0%) | 0 (0%) |

[a]IMD Quintile 1 indicates the most disadvantaged neighbourhoods and Quintile five the least disadvantaged neighbourhoods.

## Effects of lockdown on daily living during the COVID-19

**Impact of lockdown on physical activity and routine.** Older adults articulated a notable shift in their daily routines during the COVID-19 lockdown, resulting in a decline in physical activity and exercise. This transitioned towards a more sedentary lifestyle, influencing both their indoor and outdoor activities. Despite participants initial efforts to maintain outdoor physical activity, such activities diminished with the onset of adverse weather conditions and subsequent

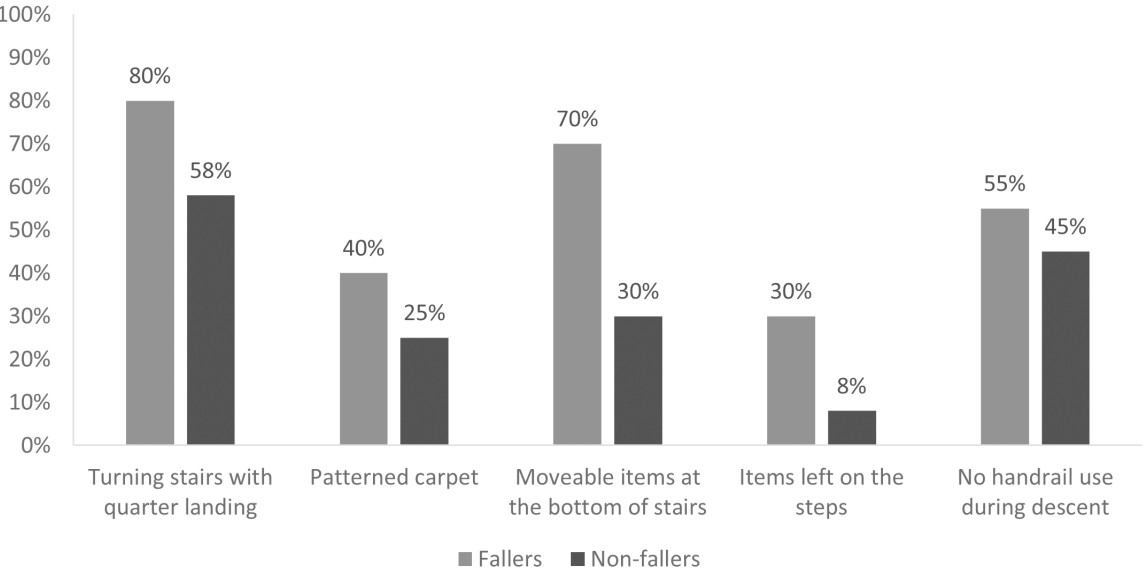

**Fig 2. Potential staircase hazards identified during home assessments.**

lockdowns. Although this observation is not particularly unusual, the previous classes or activities that would get older adults out of the house during the colder and darker months were unavailable due to the lockdown restrictions. While some attempted to shift their exercise routines to online classes as a substitute, most participants faced challenges with this transition having long term affects. The lack of physical activity during lockdown contributed to overall physical well-being concerns among participants and while only a few participants reflected on falls, the ones who did commonly expressed uncertainty about the causes of these falls. A few participants often suggested a probable link between reduced exercise during lockdown and the occurrence of their falls. This connection between sedentary behaviour and health concerns is an aspect of the broader challenges faced by older adults during the pandemic.

**Emotional and social impact of lockdowns.** With the rapid spread of the virus and limited understanding of its implications, older adults often found themselves dealing with the emotional and social repercussions of lockdown and social distancing measures. Public transportation was regarded as a potential hotspot for virus transmission due to close quarters and shared spaces, which lead to the participants' fearing leaving the house. Furthermore, participants frequently articulated feelings of isolation, frustration, and a longing for social interactions. This frustration of the loss of social engagement due to lockdown led one participant to experience a fall. This sentiment of frustration was echoed when the participants discussed how they coped with the discomfort of lockdown by immersing themselves in household projects as a distraction. The emotional complexity of living through a lockdown was shown to be a challenge. Especially when the participants spoke about major life changes, such as the loss of a spouse, the absence of familiar gatherings, and the change in the dynamics of human interaction during lockdown.

### Stair-related falls and perceived causes

**Home environment hazards.** Participants identified various elements such as the presence of a stair lift, rugs on the floor, tight turns, and lighting as potential hazards contributing to incidents and falls. For instance, the lack of a continuous banister and slippery surfaces were cited as factors in several falls. The participants' experiences underscored the importance of considering not only the visual aspects of safety features, but also their tactile properties, in assessing and addressing fall risks on stairs. Rugs, especially on wooden floors, were noted as trip hazards, and the tactile properties of

**Table 2. Themes and quotes from participants on the impact of covid-19 and falls on home stairs.**

| Non home stair fallers | Near-Fallers on Home Stairs | Home Stair Fallers |
|---|---|---|
| **Theme 1: Effects of Lockdown on Daily Living during the COVID-19** | | |
| **Impact of Lockdown on Physical Activity and Routine** | | |
| | P5: "Immediately during lockdown I was doing no exercise except for maybe an online class… COVID changed people's lifestyles, and that didn't ever get back to normal really" [Near Faller-Female5, aged 60–69]. | P7: "The first lockdown we were careful, we would just go out for an evening stroll when the weather was good…the winter times we just didn't go out… as I say we had our own in the house activities" [Stair Faller-Female7, aged 70–79]. |
| | P5: "I started to get symptoms of like a cystitis and I ended up having to have antibiotics. I was just having painful urinary tract infections because I was just sitting…I wasn't moving around enough" [Near Faller-Female5, aged 60–69]. | P7: I've only ever had the one fall on the stairs…I've just no idea really what caused that fall. But it was probably because I have been getting less exercise in that year, because that was the year wasn't it, when people didn't do anything much. [Faller-Female19, aged 70–79] |
| **Emotional and Social Impact of Lockdowns** | | |
| P4: "During COVID, we were afraid to go anywhere, to do anything, and especially travel on public transport." [Non-faller_Female4, aged 70–79]. | | P2: "It was probably because of lockdown that I fell because I was frustrated by not being able to go out and see people that I started doing a lot of projects around the house…I was manically doing things to displace discomfort and to distract myself." [Stair Faller-Female2, aged 70–79]. |
| | | P11: "I was trying to distract myself by doing something…I was doing things to the house, I was ordering stuff from Ikea…I bought myself a hammock for the garden because that's the other thing as well, there's two sides to it. You need to keep busy, but you might die soon, so you know I deserve a hammock" [Stair Faller-Female11, aged 60–69]. |
| | | P2: "I did have patches of feeling upset and desperate. I wasn't used to being by myself because my husband had died less than a year before…So you know, it was it was quite hard." [Faller-Female2, aged 70–79] |
| **Theme 2: Stair-Related Falls and Perceived Causes** | | |
| **Home Environment Hazards Leading to Falls** | | |
| | | P17: "Because there's the stair lift at the bottom, and there is a turn in the stairs to the final two. There is a Bannister rail, but it does not go around the corner. That's when it happened." [Faller-Female2, aged 70–79]. |
| | | P17: "It was wet underfoot and it was a very slippery surface. I put one foot down and it went vroom, and I fell really heavily on my back. My back went onto the nose of the other steps and I got a terrible bruise there, it really was very spectacular." [Faller-Male17, aged 70–79]. |
| | | P2: "I had two very beautiful Bokhara rugs on the wooden floor in my hall. So they were trip hazards, weren't they?... I did skate on them. If they hadn't been there, whether I would have still fallen and broke my ankle so much… and it's a tight right turn because of what I said about the very narrow hall" [Faller-Female2, aged 70–79]. |
| | | P19: "I've always had to sort of pull myself up with the banister. And I think they're a bit slippery and my right hand slid a bit." [Faller-Female19, aged 70–79]. |
| | | P7: "I can't even remember the type of stairs I have... I can't remember." [Faller-Female7, aged 70–79] |

*(Continued)*

**Table 2.** (Continued)

| Non home stair fallers | Near-Fallers on Home Stairs | Home Stair Fallers |
|---|---|---|
| | | P11: **"Interviewer:** Do you think your home stairs are safe? **Respondent:** Yes, I think I'm the element that probably isn't safe…I must stop running up and down stairs and jumping across them because you know, I can't do it anymore." [Faller-Female11, aged 60–69]. |
| **Unsafe Behaviour Leading to Falls** | | |
| | P5: "Oh it was slippers. Yeah, slipper and it was where you catch right on the edge of the step" [Near Faller-Female5, aged 60–69] | P20: "I just grabbed a load of teabags and put them in my hand...My foot just slipped and I couldn't grab hold of the rail because I had teabags in it...I would've fallen anyway even if I hadn't had the teabags because I couldn't hold on to the banister with my broken hand." [Faller-Female20, aged 80+] |
| | | P11: "I can't really tell you how it happened. I just missed it somehow. Do it every day and I tend to run up and down stairs and our stairs divide at the top, I tend to jump across. I mean it's all not recommended you know...I run up and down stairs. But there is this thing as you get older you use it or you lose it. So perhaps I should just run upstairs and not down stairs, perhaps that's… (Laughs) Mind you, I can fall upstairs as well can't I? It's like people are telling us to exercise all the time, but they don't tell us what sort of exercise, and they can't because it's very individual." [Faller-Female11, aged 60–69]. |
| | | P2: "It was the oven timer and I thought I don't want my dinner to be scorched. I just ran down the stairs…I shot down the stairs, much to my detriment because that's when the accident happened" [Faller-Female2, aged 70–79] |
| **Consequences of Falls and Injuries on Stairs** | | |
| | P5: "And then the next day I got bruises on my arm where I'd whacked against the banister" [Near Faller-Female5, aged 60–69] | P7: "I did feel stiff and sore on my shoulder, and my knee that two years previously I'd had knee replacement, hurt my elbow and the side of my head...And then I managed to turn over onto my side and crawl to the bottom of the stairs and hold on to the railing to pull myself up and sat there for a while" [Faller-Female7, aged 70–79] |
| | | P21: "I put my foot on the top step and I went up in the air and just absolutely smacked down…really cracked on my arm…And instead of landing flat on my coccyx, I landed on the one cheek really, in particular. I knew I'd really hurt myself... my backside was starting to burn and burn and burn, and swell up more and more...when I did go to hospital, to A&E, and the doctor saw me, he said, "It's a massive haematoma"" [Faller-Female21, aged 60–69] |
| | | P7: "Immediately after I was surprised. How has this happened? Why am I down here? Why am I on the floor? and then feeling stiff and sore. I sat there for a while and really it just surprised me." [Faller-Female7, aged 70–79] |
| | | P20: "I fell in the garden and broke my wrist...Then as my wrist was healing, as it was getting better...I slid down just backwards. I didn't fall backwards, I just slid down the stairs but I cracked my knee. I had two cracks in my fractured knee, two fractured ribs and of course I did hurt this wrist again...I was in hospital for 12 weeks waiting for this to heal, which took me right to Christmas Eve, I got home on Christmas Eve." [Faller-Female20, aged 80+] |
| | | P20: "Just annoyed with myself, I was just angry with myself for being so stupid...I just felt angry with myself, really. I should've been a bit more careful." [Faller-Female20, aged 80+] |

*(Continued)*

**Table 2.** (Continued)

| Non home stair fallers | Near-Fallers on Home Stairs | Home Stair Fallers |
|---|---|---|
| | | P10: "I had just come up the stairs. I was trying to carry too much… I put a bag down and it obviously was the weight factor and I started falling to one side…that did actually frighten me, and it made me think, don't be stupid, you shouldn't be doing this, but it hasn't actually stopped me" [Faller-Female10, aged 60–69] |

**Theme 3: Fall Preventative Measures and Safety Awareness**

Use of Handrails for Stability

| P6: "I do tend to use the handrails now and I probably didn't before. But is that just a thing that you get nailed at?" [Non-faller-Female6, aged 70–79] | P5: "I slipped down two but I got hold of the banister…I always put my hand down the banister rail, just automatic run my hand down the banister rail." [Near Faller-Female5, aged 60–69] | P10: "Sometimes I'm aware that I don't feel safe, and I'll grab the rail or touch the wall to make sure that I'm feeling safe." [Faller-Female10, aged 60–69] |

**Caution and Awareness for non-fallers**

| P3: "I've learned the two most dangerous places in the home is getting in and out of the bathtub to have a shower…and the stairs. So now I am aware of the stairs" [Non-faller-Male3, aged 70–79] | | |
| P5: "I be more cautious if I was carrying something down like if I'm bringing the laundry basket down" [Non-faller-Female5, aged 60–69] | | |
| P8: "The landing light is one of these ones an Energy Saver ones. So it's pretty dim…I find you when you put it on, you're just hanging around and you could be up there by the time it's lit up probably." [Non-faller-Male8, aged 70–79] | | |

**Physical Health and Fitness as Factors in Preventing Falls on Stairs**

| P12: "I fell onto the top part and rolled over, which as a runner you learn to do because you don't and a half get smacked when you hit the curb as a runner. So you start learning how to do the roll. You get scratched to death but it's saves you a lot of grief" [Non-faller-Male12, aged 70–79]. | P5: "I suppose because I'm fairly fit. So I was able to just grab it, we have a big banister rail… I've done Yoga and Pilates so my balance is better and I was able to just regain balance, that and my hands on the banister rail and being quite fit enough to grab hold of it" [Near Faller-Female5, aged 60–69] | P10: "I also have Type 2 Diabetes and I noticed it particularly in my feet more than anywhere… I go and have a massage every fortnight on my feet…So I am very conscious… And that makes me aware that I can fall down stairs". [Faller-Female10, aged 60–69] |

**Routine Adaptions after a Fall**

| | | P7: "And although I'm being careful I have to watch how I'm walking out of the conservatory where there are two ridges. And there are two steps down from the decking to the bathroom toilet. So mostly now, I would use that downstairs toilet." [Faller-Female7, aged 70–79] |
| | | P2: "Well, I'm just more cautious and more careful... I don't think I'm a cautious and careful person by nature, but I have learned to be." [Faller-Female2, aged 70–79] |
| | | P7: "Now I do hold the rail more and every time I go up or down the stairs, I count the stairs 12 up and then one to the left. I just automatically count them all now to make sure I'm not missing any, that is not my fault that I've just forgotten I'm not at the bottom. I just do that automatically now." [Faller-Female7, aged 70–79] |

*(Continued)*

**Table 2.** (Continued)

| Non home stair fallers | Near-Fallers on Home Stairs | Home Stair Fallers |
|---|---|---|
| Home Design and Safety Concerns | | |
| P6: "Sometimes I do think the landing isn't wide from the bathroom and I have thought you know, if you've got a bit confused or stumbled, you'd go straight down the stairs" [Non-faller-Female6, aged 70–79] | P5: "We could have a handrail on the wall side as well against the wall because my partner had a new hip…and one of the things is whether he'd be able to get up the stairs." [Near Faller-Female5, aged 60–69] | P2: "I was very tempted to get rid of the stair lift because it gives you more space as you're doing the tight turn. I have tried having the stairlift at the very top of the stairs. But I think that's a tad precarious because there's another tight turn. And I suppose I don't want to get rid of it because once I could use it, it was very valuable to me." [Faller-Female2, aged 70–79] |
| P22: "Interviewer: What are those barriers to growing old in this home? Non-faller: I think it's probably the… the stairs are going to be a component of it, whether it's the first component or not, I don't know… we're not immortal. So maybe the long plan should be enjoy it while we can and then look to enjoy something that might fit in better when you're a bit older" [Non-faller-Male22, aged 60–69] | | P2: "It's kind of future proofing to retain it…I suppose there is a possibility that another, an extension to the banister going round a corner might be useful, or even a vertical grab rail though I come to think about it." [Faller-Female2, aged 70–79] |
| | | P7: "At the moment coming into the front door, there's one steep step, and then another steep climb into the front porch." [Faller-Female7, aged 70–79] |
| | | P19: "It's patterned. They're all out of fashion these patterned carpets, but we like them. Everybody's got plain ones now, haven't they? We were looking for a new carpet, and there was nothing we liked. They all look the same, all grey and light grey...I don't know if it makes a difference to safety" [Faller-Female19, aged 70–79] |
| **Theme 4: Attitudes towards Ageing and Care Services** | | |
| **Financial Considerations and Concerns in Ageing** | | |
| P8: "The only thing I thought of in that respect would be a stairlift eventually. And I think there about £6000…I think if we need really needed one then we manage it somehow other than that, it will be a downstairs toilet. Which was one of the reasons why we left the other place and moved." [Non-faller-Male8, aged 70–79] | | P10: "I know there are some beautiful care homes, but then they cost beautiful amounts of money" [Faller-Female10, aged 60–69] |
| P9: "Interviewer: Do you feel you're financially able to upgrade your home as you age? Participant: It's getting harder" [Non-faller-Female9, aged 60–69] | | P20: "I'm getting frustrated because I can't see and I can't hear properly, that is getting me down a bit, that's making me feel like I have got to have somebody to help me." [Faller-Female20, aged 80+] |
| P4: "We didn't know how much income we had…and with this energy crisis coming I'm paranoid about leaving lights on and turning lights on unnecessarily." [Non-faller-Female4, aged 70–79] | | P7: "It's reluctance to admit, I'm beginning to get a bit older and beginning to need these things…Because older people usually hold on to both rails as they're trying to get themselves up the stairs. And I feel at the moment, I can still cope with one. But then as I say this, Falls Prevention Team have tried to persuade me that it's my own benefit to have the other one." [Faller-Female7, aged 70–79] |

*(Continued)*

**Table 2.** (Continued)

| Non home stair fallers | Near-Fallers on Home Stairs | Home Stair Fallers |
|---|---|---|
| P8: "You lose so much independence. They are regimented up to a point which they probably have to be but you can't do anything. I mean, we've got so many things we can do here the garden, you couldn't do that. The amount of reading that we do. I mean, I've got several hobbies as well. I couldn't do those. There would be nowhere to put anything that we use. So just the massive loss of independence from ability to do what you wanted and when you wanted." [Non-faller-Male8, aged 70–79] | | |
| **Seeking and Receiving Care Services** | | |
| | | P7: "Unfortunately, I had one or two falls...that's when the Falls Prevention Team became involved with me. And they helped the assistants, they put a rail down there for me to get down the decking easier. And a rail in the bathroom, and one by the door. So that's when that service came in. And that's how I've got those appliances "[Faller-Female7, aged 70–79] |
| | | P19 "I've got more active with doing all the training with the dog. And so, I think it's done me good, because I was really starting to see stuff. And having problems with arthritis in my knees and feet and back and so on, and actually, having to keep moving around, I keep going every day out and I felt better in the last few weeks." [Faller-Female19, aged 70–79] |
| | | P21: "I hadn't been hospital when I first did it, because of lockdown, because of the pressure on the NHS and everything, and I was like, I'll be all right" [Faller-Female21, aged 60–69] |
| **Emotional Attachment to the Home and Reluctance to Move into Care Homes** | | |
| P8: "I would be extremely reluctant to do anything like that. Yeah, very, very, very reluctant" [Non-faller-Male8, aged 70–79] | P5: "Like a real feeling of security, I suppose. It's always a thing where or when you come home from work. It's nice to close the door and close the curtains and it's, oh, you can sort of breath out and relax." [Near Faller-Female5, aged 60–69] | P2: "I was offered a care package. And for two days carer's did come in. But that made me feel very concerned because they would be going to lots of places…It's hard, isn't it, to remember how scared you were during the first lockdown... I think I thought if I put my nose through the front door something awful would happen" [Faller-Female2, aged 70–79]. |
| P12: "Well I don't think you get much care and I think once you get to certain age like people seem to think you're invisible, especially in care homes. So you know, I'd rather snuff it at home." [Non-faller-Male12, aged 70–79] | | P10: "I would rather go to Dignitas or have something by that time, hopefully in this country where you can say right, I would just want to end it." [Faller-Female10, aged 60–69] |

safety features like bannisters were highlighted as important for preventing falls. However, many participants struggled to recall what their stairs looked like or acknowledge their stairs as the reason for their falls and injuries.

**Behavioural hazards.** Some respondents attributed their falls to their own actions, such as running or jumping on the stairs. These responses suggest a behavioural aspect linked to the falls experienced rather than the design of the stairs being considered the sole causal factor. Participants frequently discussed various situations where unsafe behaviours, such as carrying too many items or heavy loads, wearing inappropriate footwear, or rushing downstairs, led to falls and

injuries. The narratives of these stair fallers reflected a mix of casual attitudes towards stair use, a lack of awareness about the risks involved, and the absence of a clear recollection of the fall adds an element of unpredictability to their behaviour. These incidents illustrate how unsafe behaviours on stairs can lead to severe outcomes and underscore the importance of addressing behavioural risks to prevent stair-related falls among older adults.

**Consequences of falls and injuries on stairs.** Older adults reported about the physical and emotional tolls of stair-related falls. Participants frequently described feeling stiff, sore, and bruised after their falls, with some sustaining severe injuries such as cracked arms, bruised limbs, and haematomas. Not only were these consequences echoed by the older adults who had experienced a stair fall, but we also found that an injury was sustained from an older adult who experienced a near fall. The immediate reactions to these falls were often marked by surprise, confusion, and fear, leading them to re-evaluate their actions and behaviours on stairs. The physical impact of these falls included mobility limitations and prolonged recovery periods, significantly affecting their daily lives and activities. In some cases, initial falls led to additional, more serious falls, resulting in extended hospital stays and further complications. In addition to physical consequences of falls, emotional responses were common among participants, reflecting self-blame and regret regarding the circumstances of their falls. Despite recognising the risks and emotional impact, some participants admitted to continuing unsafe behaviours after experiencing a fall, suggesting challenges in modifying behaviour. These findings present a multifaceted story of falls, injuries, emotional responses, and the lasting impact on the participant's mobility and independence.

## Fall preventative measures and safety awareness

**Use of handrails for stability.** Participants widely reported that handrails played a pivotal role in their safety and stability when navigating stairs and often emphasised how they'd made conscious efforts in using the handrails to prevent falls. Some participants came to realise that they adapted their behaviour as they aged, becoming more inclined to use handrails for stair safety. This observation highlights a common trend among older adults to prioritise safety measures such as handrail usage as they navigate stairs, reflecting a broader recognition of the importance of injury prevention and fall mitigation strategies with age. Additionally, the role of the handrail was not just seen as a supportive and preventative measure by older adults but actually helped them mitigate the impact of a fall. The willingness of older adults to adapt and utilise handrails when feeling unsteady was captured by most respondents. Despite varying levels of autonomy in using handrails, participants actively sought support from them to ensure a sense of safety during stair navigation.

**Caution and awareness for non-fallers.** The majority of participants who did not experience a stair fall reported on their conscious efforts and awareness of potential risks and hazards when using the stairs, as well as their proactive measures to ensure safety. Respondents often emphasised their increased level of concern and adaptive behaviours to reduce the potential for falls, particularly when carrying items up or down the stairs. Additionally, participants discussed how dim lighting affected their ability to navigate the stairs safely. The consistent mention of being more "cautious" among all non-faller participants indicates a shared recognition of the potential risks associated with multitasking while on the stairs.

**Physical health and fitness as factors in preventing falls on stairs.** Older adults highlighted the role of physical health and fitness in preventing falls on stairs. They often suggested that the older adults who were physically fit might find it easier to maintain balance and be more stable when using stairs, potentially reducing the risk of falls. Participants who engaged in regular exercise, such as yoga, Pilates, or running, reported better balance and quicker reactions to potential hazards, attributing their enhanced stability and fall prevention to their fitness levels. While the majority of participants highlighted the importance of physical fitness, respondents who had experienced a stair fall acknowledged the potential impact of underlying health conditions, on stair safety. The older adults who maintained good physical fitness tended to feel more confident and secure when using stairs, while those with underlying health conditions were more aware of the potential risks and took preventive measures to address them. This reflects a broader awareness among older adults of the potential impact of health conditions on their safety and mobility.

**Routine adaptations after a fall.** Older adults who had experienced a fall reported that they adapted their routines to prioritise safety, by becoming more vigilant about potential hazards in their environment, such as uneven surfaces and steps, and adjusted their behaviour to mitigate and avoid these risks. For example, instead of navigating hazardous routes, they chose safer alternatives, such as using facilities on a single level. This change in routine highlights the reactive measures taken by older adults to prevent future falls, showcasing their heightened caution and deliberate actions after a fall. Moreover, participants reflected on the psychological transformation they've undergone, acknowledging that they became more cautious and careful as a result of their experiences. Additionally, older adults frequently delved into their specific and individualised routines that they had developed to enhance stair safety after a fall. Although older adults frequently reported on their individualised routines, it conveyed an overall responsive effort rather than proactive measure to prioritise safety and prevent a reoccurring stair fall.

**Home design and safety concerns.** Respondents often expressed safety concerns regarding their living environment. They often indicated a heightened awareness of the potential dangers associated with the specific architectural features of their home, such as narrow landings and tight staircases. It was common for the respondents to discuss their narrow staircase. However, only a few participants discussed their narrow staircase in the context of the consideration and decision-making process regarding the stair lift in their home. Furthermore, the idea of future-proofing their homes was mentioned by multiple respondents, indicating a shared recognition of the importance of planning ahead for ageing-related changes. Older adults often offered practical solutions, such as installing additional handrails, to enhance safety and support during stair use. Concerns about architectural features, like steep entryway steps, were also raised, and older adults emphasised the need for more accessible designs. When asked about the barriers to ageing in their current homes, many older adults identified stairs as a significant challenge, although not necessarily the most immediate concern. They acknowledged the inevitability of ageing which prompted evaluations of their living situations.

## Attitudes towards ageing and care services

**Financial considerations and concerns in ageing.** Participants discussed an array of expenses linked to ageing. A focal aspect of financial consideration revolved around the expenses tied to adapting home environments for safety and accessibility. While some participants expressed determination to manage the financial costs, others reflected on their financial ability to upgrade their homes as they age differently. Many expressed concerns about the affordability of such upgrades when attempting to make their homes more age friendly. Beyond home modifications, respondents also touched upon the financial considerations associated with potential future care needs. The prospect of residing in care homes involves significant financial planning, and often participants expressed concerns about these potential costs. This sheds light on the intricate financial landscape that older adults must navigate as they contemplate their future living arrangements. Additionally, the financial considerations extended to day-to-day expenses, including energy costs. Economic challenges, such as increases in energy costs or uncertainties about income, were mentioned, highlighting the impact of broader economic factors on older adults' daily decisions and behaviours. Moreover, respondents often expressed their emotional and practical aspects of ageing, and the desire to remain independent. Frustration and feelings of dependence on others for assistance were discussed, reflecting the challenges older adults face in overcoming sensory limitations alone. This tension between the desire for many older adults to maintain independence, the acknowledgment of the potential benefits of safety measures and the implementation of these safety features, was resounding. Many participants emphasised the importance of preserving their ability to make choices and decisions.

**Seeking and receiving care services.** A few participants described their experiences with the receiving care services, indicating their active engagement in efforts to mitigate the risk of falls. Likewise, receiving care services, such as accessing a guide dog had benefits of increasing activity levels and positive effects on respondents' well-being. However, access to and seeking care during the COVID-19 pandemic was somewhat challenging for the respondents. Among the respondents who received care services during the pandemic, there was shared a sense of risk and anxiety associated

with any form of external contact, due to the fear of contracting COVID-19. Overall, these findings reflect older adults' proactive approach to maintaining their health and independence by seeking and receiving different types of care and assistance as they age. It also highlights the impact of the fear of contracting COVID-19 had on respondents receiving care services, despite their potential need for assistance.

**Emotional attachment to the home and reluctance to move into care homes.** The emotional attachment to the home and reluctance to move into care homes were consistently reported. Many of the older adults included sentiments of comfort, security, and the significance of homeownership. Participants often reflected on the idea that their homes are places with deep emotional and practical importance in their lives. Moreover, the respondents often expressed strong reservations and aversion to the idea of leaving their current homes to live in care homes or assisted living facilities. The respondents' reservations about care homes stemmed from concerns about the quality of care, the potential loss of independence, and the regimented nature of such facilities. Furthermore, participants expressed a preference for facing end-of-life issues in their familiar home environment rather than moving into care homes. This was a common response to the idea of moving into a care home. Older adults often expressed reluctance to consider alternatives such as moving into care homes, emphasising the importance of maintaining control over their own lives and decisions, even in matters related to the end of life.

## Discussion

This mixed-methods study provides insight into older adults' experiences of independent living and home stair falls during and after the COVID-19 pandemic. Although the COVID-19 lockdown provided an opportunity to explore fall risk and stair safety, our findings extend beyond the context of the COVID-19 pandemic. They highlight a broader gap between older adults' perceptions of staircase safety and the actual risk factors present in their homes. We identified four overarching themes common across older adults' living independently: effects of lockdown on daily living during the COVID-19 pandemic; stair-related falls and perceived causes; falls preventative measures and safety awareness; and attitudes towards ageing and care services. The journey to attain and sustain older adults' independence within their homes is intricate, requiring not only increased awareness, education, and modifications in the physical environment, but also considerations of behavioural and situational factors.

Every single stair faller perceived their stairs as safe, despite a substantial proportion (40%) of them having staircases that did not meet government safety guidelines. Reinforcing these findings, the English Housing Survey Headline Report states that only 9% of homes met the basic standard of accessibility set by government [18]. This discrepancy may be due to the UK's growing older population and the fact that the UK possesses one of the oldest housing stocks in Europe [4,5], making it challenging to meet modern safety standards. Interestingly, our measurements of stair dimensions failed to distinguish between those who had experienced home stair falls and those who had not, indicating that both groups equally did not meet the safety standards. This suggests that behaviour and décor choices may play a pivotal role in home stair safety. Behaviours such as leaving items on the stairs or neglecting to use handrails and décor choices such as patterned carpet were identified as influential factors in home stair safety.

However, this finding should not downplay the inherent risk, as adhering to safety standards, specifically those set by the British Standard BS5395−1, could potentially reduce the risk of falls by 60% [15]. Our interviews supported the importance of adhering to safety standards, especially the provision of handrails, with several instances where their use prevented more severe falls. While the structural transformation of existing stairs (pitch, rise and going) poses challenges, updating building regulations for new constructions emerges as a proactive preventive strategy [19]. For homes with non-compliant staircases, interventions targeting elements of safety standards, such as slip resistance, stair coverings, and the provision of handrails, can be effective [19,20].

Regarding stair coverings, participants recognised that loose or worn carpets could be a potential hazard. However, 90% of participants who had patterned carpet did not perceive them as a risk, indicating a lack of awareness of the

increased fall risks. Surface colour and/or pattern can affect visually locating step edges or judging distances, reduce confidence and increase anxiety for stair negotiation [21,22]. Educational initiatives that inform older adults about the role of behaviour and décor choices in home stair safety are imperative [23]. Increasing the older adults and their carer's ability to identify hazards, with the knowledge of how to access support for home adaptions can minimise the stair fall risk and risky behaviours [23]. This underscores the need for targeted interventions to modify behaviour and promote safer stair practices among older adults.

Participants also reflected on unsafe behaviours that resulted in falls and injuries on stairs, such as misjudging step edges, carrying items, or rushing. These falls had considerable impact on their well-being, with consequences such as broken bones, discomfort, and increased fear when navigating their homes. This prompted a reactive response rather than preventative. Reduced confidence and increased anxiety have been linked to changes in visuomotor behaviour during obstacle negotiation, such as decreased walking speed [22,24] and reduced visibility [25]. Anxiety may cause older adults to fixate on future hazards earlier, inducing an attentional bias toward environmental features perceived as more threatening [22,26]. This premature transferral of gaze from a stepping target such as the next step to future obstacles such as a moveable item left on the top of the stairs has been shown to reduce stepping accuracy and increase foot placement variability, which could exacerbate fall risk [22,26]. Therefore, prevention strategies, such as adaptions to the home and education are needed before a fall occurs [2]. Addressing unsafe behaviours can effectively reduce the overall risk of stair falls among older adults [27].

However, our study only captured a brief understanding of the participants behaviour negotiating stairs. We understand that factors such as patterned carpet, items left on the steps or having turning stairs with quarter landings (90°/180° turn to a second flight) may also amplify fall risk. Employing cautious stepping technique might mitigate the likelihood of a stair fall, potentially compensating for reduced functional capabilities or inadequate safety standards [27]. Further research is required to explore the behavioural aspects of stair safety to understand additional perceptions and differences in stepping techniques between fallers and non-fallers, informing effective fall prevention interventions.

During interviews, participants discussed their safety concerns and approach to 'future proofing' their homes, emphasising the significance of handrails, lighting, and tight turns on stairs. Financial challenges were significant, as participants often could not afford home upgrades and avoided using lights and heating to reduce costs. Poor lighting can exacerbate stair fall risks, influencing confidence, anxiety, and dynamic balance [28]. Cold homes can also have a severe impact on both physical health (such as respiratory or circulator problems) and mental health, making negotiating stairs a challenging and riskier task [19]. It is clear that there is a huge and urgent need to adapt UK homes. The English Housing Survey (2021) indicates that around 820,000 homes have issues like damp or mould growth, and this is seen as a more concerning issue than adapting their homes for future use or making them more energy efficient [3,18]. Previous research and our findings show that that people cannot afford to make adaptions to their home to ensure their safety and prevent care needs [3,18,29]. Additionally, the high costs of quality care homes underscore the financial burdens and uncertainties older adults and their families confront when contemplating future care options. It is clear that home adaptions can play a key role in supporting older adults to live independently for longer, equally, it can act as a preventive measure against the escalation of care requirements as a result of stair falls and accidents within the home [3].

When we factor COVID-19 lockdown into this, the risk of falls and injuries is amplified beyond the usual levels as prolonged period of home confinement during the pandemic may have heightened exposure to these potentially 'unsafe' home [8–10]. Lockdown measures disrupted regular exercise routines, leading to a decline in physical movement. This has potentially affected the muscle strength and balance necessary for safe stair negotiation [23,28] and contributed to deconditioning and diminished physical fitness, increasing the susceptibility to stair falls, especially for older adults [8]. The long-lasting effects of the closure of organised activities and services were evident as individuals faced difficulties returning to their pre-lockdown routines. These highlight the need for implementation of effective exercise programmes for older adults especially during periods of self-isolation and home-confinement [23].

The varied and sometimes contradictory participant narratives reflect the complex relation between perceived and actual fall risk, and the individual ways older adults respond to the risk. For example some participants actively use handrails or modify behaviour, whereas others express reluctance or overconfidence in their mobility. In addition, some acknowledge environmental hazards, while others attribute falls to personal behaviour or fitness levels. In regards to lockdown, some experienced frustration leading to risky behaviour, whereas others adapted cautiously. These differences highlight how individual experiences that are shaped by physical environment, personal attitudes and situational factors contribute to different perceptions of stair safety, risk-taking behaviours, and fall prevention strategies. This variability highlights the need for flexible, person-centred approaches to fall prevention. The implications of this study extend beyond the COVID-19 pandemic, as the environmental and behavioural factors compromising stair safety that were identified will likely remain until specific actions are taken to mitigate stair fall risk. This study revealed a previously unacknowledged issue: many older adults' homes fail to meet government guidelines for stair safety, leaving staircases with numerous unaddressed hazards. While COVID-19 provided an opportunity to examine these environments due to increased time spent at home, the associated rise in loneliness and decline in physical activity among older adults [8–11] emphasises the ongoing importance of addressing home safety beyond the pandemic context. This calls for further research and educational interventions to raise awareness of home hazards and promote fall prevention strategies.

## Strengths and limitations

The findings of our study emphasise the importance of fall prevention teams, particularly occupational therapists (OTs) conducting ongoing objective assessments of home environments, to identify potential risks for stair falls specifically. In the UK, fall prevention teams provide assessment and rehabilitation for adults who have fallen. This multidisciplinary team, which includes physiotherapists, occupational therapists, and podiatrists, works with patients in their homes and communities to restore independence, improve confidence, and reduce hospital admissions [30]. The involvement of OTs usually takes place post-fall (intervention stage) due to lack of resources to assess fall risk and intervene accordingly before the occurrence of the first fall. However, given that a fall is on its own a risk factor for consequent falls [2], the fall prevention team should be involved earlier, to prevent the first fall in those identified as high-risk older adults. Implementing multicomponent interventions such as community fostering, housing adaptions, and physical activity recommendations could all potentially help prevent home stair falls for older adults [23], and need to be addressed in future research.

We acknowledge the study has certain limitations. Firstly, we did not collect detailed information on ambulatory impairments or the use of mobility aids such as canes or walkers. The use of walking aids has been identified as a potential risk factor for future falls among older adults [31], which may have influenced our findings. Another limitation may have been recall bias. As individuals age, there can be variations in the accuracy and completeness of their recollections, potentially leading to recall bias [32]. The ability to recall specific details, especially those related to past events or experiences during the COVID-19 pandemic, may be influenced by factors such as cognitive function, health conditions, and the emotional significance of the memories [33–36]. Despite efforts to minimise this limitation through careful interview techniques and validation of information, it is crucial to acknowledge that differences in recall abilities among participants might have impacted the comprehensiveness and accuracy of the collected data [37]. While these insights provide valuable qualitative data, future research may benefit from additional methods, such as cross-referencing participant accounts with records from Hospital Episode Statistics (HES) and general practitioner data records on falls, to further enhance the robustness of data gathered from older adults [37]. In addition to recall bias, this study may also be subject to self-selection bias, as individuals who chose to participate might have had stronger opinions or experiences related to the topic, potentially limiting the generalisability of the findings. An additional limitation is the homogeneity of our sample, which may impact the generalisability of our findings and that the representation of diverse communities may not have been captured. The study included participants that were from a White ethnic background, living in more affluent areas (IMD Quintiles 3, 4, and 5) and mostly homeowners. This limited representation may not fully capture disparities present in

other populations, including ethnic minorities and individuals from lower-income backgrounds. Inclusivity and representation in studies is vital for recognising and addressing the needs of diverse communities and socio-economically disadvantaged areas [10,38]. Therefore, solutions may be less effective, further continuing health disparities, and future research aiming to address this issue is needed.

## Conclusion

Our study illustrates the gap between what older adults perceive as safe staircases and the actual risk factors present in their own homes. The COVID-19 lockdown provided an opportunity to explore fall risk and stair safety, however, our findings expand beyond the context of the COVID-19 pandemic. Findings highlight the desire of wishing to remain living as independently as possible in their own home, and how stair falls can deeply affect the way older adults navigate their homes. Findings also highlights the importance of conducting objective assessments and emphasise the role behavioural factors may play in stair safety. Stair fall prevention strategies should take a multifactorial approach, with education, home adaptions, community-based interventions and allowing for collaborative support for ageing in place. Installing safety features and implementation of educational practical interventions can help minimise fall risk and enhance the safety of older adults.

## Supporting information

**S1 Appendix. Semi-structured topic guide used in interviews.**
(DOCX)

**S2 Appendix. Stair assessment protocol for evaluating staircase safety.**
(DOCX)

## Acknowledgments

We sincerely thank the older adults who participated in this study for their time and valuable insights, which were crucial to this research. We also extend our gratitude to the organisations and forums that supported this work, including The Orrell Trust, Sefton Older Persons Forum, the NIHR Applied Research Collaboration North West Coast Public Advisor Forum, and the Women's Club Forum. Their contributions and collaboration were invaluable in reaching and engaging with the participant community. We acknowledge the National Institute for Health and Care Research Applied Research Collaboration North West Coast (ARC NWC) for funding this research as part of a PhD studentship. Finally, we acknowledge the institutional support provided by Liverpool John Moores University (LJMU), which enabled the successful completion of this study.

## Author contributions

**Conceptualization:** Emily Wharton, Constantinos Maganaris, Richard J. Foster, Thomas O'Brien, Clarissa Giebel, Asan Akpan, Mike Roys.

**Data curation:** Emily Wharton, Constantinos Maganaris.

**Formal analysis:** Emily Wharton, Constantinos Maganaris, Richard J. Foster, Thomas O'Brien, Clarissa Giebel, Asan Akpan, Mike Roys.

**Funding acquisition:** Emily Wharton.

**Investigation:** Emily Wharton.

**Methodology:** Emily Wharton, Constantinos Maganaris, Richard J. Foster, Thomas O'Brien, Clarissa Giebel, Justine Shenton, Asan Akpan, Mike Roys.

**Project administration:** Emily Wharton.

**Resources:** Emily Wharton, Constantinos Maganaris, Justine Shenton, Asan Akpan, Avril Mills, Mike Roys.

**Software:** Emily Wharton.

**Supervision:** Emily Wharton, Constantinos Maganaris, Richard J. Foster, Thomas O'Brien, Clarissa Giebel.

**Validation:** Emily Wharton.

**Visualization:** Emily Wharton, Avril Mills.

**Writing – original draft:** Emily Wharton.

**Writing – review & editing:** Emily Wharton, Constantinos Maganaris, Richard J. Foster, Thomas O'Brien, Clarissa Giebel.

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
