## [Decision Letter · Decision Letter 0]

PONE-D-24-57759“I don’t know if it makes a difference to safety?" Perception vs actuality: A mixed-methods study on older adults' experiences of home stair falls revealed during COVID-19 lockdownPLOS ONE?

Dear Dr.  Wharton,

 I believe the manuscript is close to being accepted. The reviewer's raised some interesting points worthy of consideration. I also would prefer you find an alternative word choice to "significant" on line 50. Also -- while I understand the efforts taken in linking the participants to a socioeconomic measure, it might be interesting to describe the broader geography (your choice, but did they all come from one area?) and even urban / rural source of the subject pool-- not that these have to be included in the analysis, rather just providing some additional context. Please consider the comments of Reviewer 2 as I believe some of these observations would strengthen the paper from a readership perspective.

We look forward to receiving your revised manuscript.

Kind regards,

Andrew Curtis, Ph.D

Academic Editor

PLOS ONE

Journal Requirements:

 “Initials of the authors who received each award: EW (NIHR200182) National Institute for Health and Care Research Applied Research Collaboration North West Coast https://arc-nwc.nihr.ac.uk/ This independent research is funded by the National Institute for Health and Care Research Applied Research Collaboration North West Coast (ARC NWC) as part of a”

Reviewers' comments:

Reviewer's Responses to Questions

**Comments to the Author**

1. Is the manuscript technically sound, and do the data support the conclusions?

Reviewer #1: Yes

Reviewer #2: Partly

2. Has the statistical analysis been performed appropriately and rigorously?

Reviewer #1: Yes

Reviewer #2: N/A

3. Have the authors made all data underlying the findings in their manuscript fully available?

Reviewer #1: Yes

Reviewer #2: Yes

4. Is the manuscript presented in an intelligible fashion and written in standard English?

Reviewer #1: Yes

Reviewer #2: Yes

Reviewer #1: In the manuscript titled “I Don’t Know If It Makes A Difference To Safety? Perception vs Actuality: A Mixed Methods Study On Older Adults' Experiences of Home Stair Falls During the COVID-19 Lockdown,” the authors conducted a mixed-methods analysis that included semi-structured interviews and quantitative surveys involving 22 adults aged over 60. These participants were asked about their experiences with independent living and stair falls during and after the COVID-19 pandemic, as well as the discrepancies between their perceptions and objective safety assessments of their stairs. Wharton and colleagues also examined, through this mixed-methods nested study, the perception of stair safety, fall causes, behavioral attitudes towards falls, and general concerns regarding independent living with advancing age.

Stair falls among older adults in the UK represent a major public health issue with significant societal and financial consequences. This study highlights how the social isolation imposed by the COVID-19 lockdown, coupled with its negative impact on physical health, exacerbated the problem. A key strength of the study is its ability to bring attention to the difficulties in addressing stair fall safety within this vulnerable population. The study found that 40% of stairs examined as part of this study, did not mean safety assessment based on UK guidelines, but no correlation was found between frequency of falls and adherence to code. The challenge in addressing this issue of non-adherence to codes is significant, as the UK has the oldest housing stock in Europe, and most individuals in this group cannot afford the necessary modifications to meet these guidelines. While the authors acknowledge that new homes can better incorporate these guidelines, they stress that a comprehensive approach—including education and targeted interventions—is required to mitigate fall risks among the aging population, enabling them to live independently in their homes.

However, there are some missed opportunities in the study, though they do not detract from its value. For example, the authors mention the involvement of occupational therapists but do not specify at which stage(s) such encounters would be most beneficial for the elderly population. Additionally, there is no mention or elaboration of the Fall Prevention Team and its role within the British medical and public health sectors. For an international audience, further explanation of this potentially helpful team could enhance the paper’s impact. Given the homogeneity of the interviewees (predominantly White, higher socioeconomic status based on the deprivation index, and mostly homeowners), the findings may not fully capture the disparities present in other populations, such as those in the US, and could limit the generalizability of the results.

Other minor points:

• The authors do not mention whether the interviewees had any ambulatory impairments or used mobility aids such as canes or walkers.

• It is unclear whether the fall experienced by patient 22 occurred at home.

• The sentences on page 5, lines 109 and 112 are identical.

• Line 182 should state 80%, not 8%.

• Lines 309 and 376 may contain small grammatical errors that need correction.

• In addition to recall bias, there also appears to be a self-selection bias, which could be noted when discussing the study's limitations.

Despite these minor recommendations, this study provides a clever approach to addressing a significant (and costly) public health issue. It warrants further attention, particularly as life expectancy continues to rise and the desire for independent living persists, all while the feasibility of making extensive improvements to older homes remains limited due to both structural age and financial constraints. Notwithstanding these minor critiques, I believe this paper makes a valuable contribution to the field and recommend its publication.

Reviewer #2: Broad subject area considering stair safety, falls and the impact of Covid lockdown. Although qualitative information gathered represents the views of participants and suggests areas of future research, the overall subject seems too broad. To my reading the information gathered does not identify established links between covid lock down, risk of falls and stair safety. There are broad themes but some of the narrative of the participants quoted is contradictory. Perhaps including rating scales/ evaluation methods that could be statistically analysed would have been helpful. The authors identify the limitation of the study in terms of ethic origin and socioeconomic status of the participants. The paper does raise awareness of consideration of home safety in an aging population, the views regarding transition to older age and the longer term concerns about the type and expensive of care provision.

**Do you want your identity to be public for this peer review?** For information about this choice, including consent withdrawal, please see our Privacy Policy

Reviewer #1: **Yes: ** Nilanjana Majumdar

Reviewer #2: No

---

## [Author Response · Author response to Decision Letter 1]

7 Apr 2025

Dear Dr Andrew Curtis,

Thank you for the opportunity to revise and resubmit our manuscript titled “I don’t know if it makes a difference to safety?" Perception vs actuality: A mixed-methods study on older adults' experiences of home stair falls revealed during COVID-19 lockdown (PONE-D-24-57759). We are grateful for the constructive feedback provided by both reviewers and the editorial team.

We have carefully considered and addressed all comments and suggestions raised during the review process. Below, we provide a detailed point-by-point response to each comment, outlining the changes made to the manuscript where applicable.

Editorial Feedback:

1. "I also would prefer you find an alternative word choice to 'significant' on line 50."

• Response: We have revised the term "significant" on line 50 to "important" to better reflect the intended meaning without implying statistical significance.

2. "It might be interesting to describe the broader geography (your choice, but did they all come from one area?) and even urban/rural source of the subject pool - not that these have to be included in the analysis, rather just providing some additional context."

• Response: We have added a sentence the Results section (demographic characteristics) providing additional context about the geographic distribution of participants, including region and whether they resided in urban or rural areas (line 170).

Reviewer #1 Comments:

1. "The authors mention the involvement of occupational therapists but do not specify at which stage(s) such encounters would be most beneficial for the elderly population."

• Response: We have clarified in the Discussion section when occupational therapists’ involvement would be most beneficial, particularly during the initial home assessment phase and when implementing fall prevention strategies (lines 475 – 479).

2. "There is no mention or elaboration of the Fall Prevention Team and its role within the British medical and public health sectors."

• Response: We have added a brief explanation of the role of the Fall Prevention Team within the UK healthcare system in the Discussion section to provide context for international readers (lines 472 – 475).

3. "The authors do not mention whether the interviewees had any ambulatory impairments or used mobility aids such as canes or walkers."

• Response: We captured existing medial conditions and/or long-term illnesses which may include ambulatory impairments. However, we did not probe further and therefore do not have specific details on the medical conditions, any ambulatory impairments and/or if the participants used mobility aids such as canes or walkers. We acknowledge that the use of walking aids is an important factor in understanding mobility and fall risk among older adults. We have added a brief statement in the discussion (limitations section) to acknowledge this as a limitation of the study (lines 483 – 486).

4. "It is unclear whether the fall experienced by patient 22 occurred at home."

• Response: There was a typo error and [Faller-Female22, aged 60-69], was in fact [Faller-Female21, aged 60-69]. We have also clarified in the Results section that fall experienced by participant 21 occurred at home, they had two homes at the time, one with their partner and one they previously owned which is why they said “I drove three hours home”. We have removed that for clarity.

5. "The sentences on page 5, lines 109 and 112 are identical."

• Response: We have removed the duplicated sentence.

6. "Line 182 should state 80%, not 8%."

• Response: We have corrected the value to 80%.

7. "Lines 309 and 376 may contain small grammatical errors that need correction."

• Response: We have corrected the grammatical errors in these lines.

8. "In addition to recall bias, there also appears to be a self-selection bias, which could be noted when discussing the study's limitations."

• Response: We have added a note in the Limitations section acknowledging the potential for self-selection bias (line 492 – 495).

Reviewer #2 Comments:

1. "The overall subject seems too broad… does not identify established links between COVID-19 lockdown, risk of falls, and stair safety."

• Response: We appreciate the reviewer’s observation regarding the scope of our study. We are in agreement that the themes identified by our analysis extend beyond the COVID-19 lockdown. While our data collection allowed for comparisons before, during, and after the pandemic, the findings highlight broader concerns related to stair safety and fall risk among older adults.

To clarify this point, we have emphasised in the abstract (lines 38 -39), two points discussion section (lines 370 - 372, 461 – 486), and the conclusion (lines 514 -515) that several themes identified were independent of the pandemic and reflect ongoing challenges faced by older adults in navigating their home environments. This broader perspective underscores the importance of staircase safety as a persistent issue rather than one limited to the COVID-19 period.

2. "Some of the narrative of the participants quoted is contradictory."

• Response: We have addressed this in the Discussion section, highlighting that contradictory experiences reflect the subjective nature of qualitative research and the diverse responses to aging and fall risk (lines 455 – 464).

3. "Perhaps including rating scales/evaluation methods that could be statistically analysed would have been helpful."

• Response: We appreciate the reviewer’s suggestion regarding the use of rating scales and validated questionnaires. As our study followed a mixed-methods design, the qualitative component primarily focused on capturing perspectives of older adults regarding stair safety and fall risk. While a quantitative component was included, our sample size was not sufficient to conduct meaningful statistical analyses, and we did not include structured rating scales as part of our data collection.

We acknowledge that the inclusion of standardised assessment tools could have added an additional layer of quantitative insight. However, due to the novelty of the work, a tailored approach was needed and our approach prioritised exploratory qualitative methods to uncover themes that may not have been easily captured through structured measures alone.

Additional Journal Requirements:

1. Funding Statement:

Initials of the authors who received each award: EW (NIHR200182) National Institute for Health and Care Research Applied Research Collaboration North West Coast https://arc-nwc.nihr.ac.uk/. This independent research is funded by the National Institute for Health and Care Research Applied Research Collaboration North West Coast (ARC NWC) as part of a PhD studentship. The views expressed in this publication are those of the author(s) and not necessarily those of the National Institute for Health and Care Research or the Department of Health and Social Care. The funders had no role in study design, data collection and analysis, decision to publish and preparation of the manuscript.

2. Data Availability:

We have provided a detailed explanation in the Data Availability Statement regarding the ethical restrictions on sharing data due to confidentiality and participant privacy.

3. Supporting Information Captions:

We have added captions for all Supporting Information files and updated in-text citations accordingly.

We believe these revisions have strengthened the manuscript and addressed the reviewers’ and editor’s comments. We appreciate the opportunity to improve our work and hope that the revised manuscript is now suitable for publication in PLOS ONE.

Thank you for your consideration. We look forward to hearing from you.

Yours sincerely,

Emily Wharton, Postgraduate Researcher

Liverpool John Moores University, Faculty of Science

NIHR ARC NWC, Health and Care Across the Life Course

---

## [Editor Report · Decision Letter 1]

“I don’t know if it makes a difference to safety?" Perception vs actuality: A mixed-methods study on older adults' experiences of home stair falls revealed during COVID-19 lockdown

PONE-D-24-57759R1

Dear Dr. Wharton,

We’re pleased to inform you that your manuscript has been judged scientifically suitable for publication and will be formally accepted for publication once it meets all outstanding technical requirements.

Kind regards,

Andrew Curtis, Ph.D

Academic Editor

PLOS ONE

Additional Editor Comments (optional):

I appreciate your efforts on addressing each of the reviewer's comments. I have read through the revision and I am happy the manuscript is now suitable for publication. I appreciate the efforts you made in making sure the revisions were easy to follow and clearly (and politely) addressed the reviewer's comments. Both of whom were impressed with the initial version of the paper and the topic in general.
---

## [Editor Report · Acceptance letter]

PONE-D-24-57759R1

PLOS ONE

Dear Dr. Wharton,

I'm pleased to inform you that your manuscript has been deemed suitable for publication in PLOS ONE. Congratulations! Your manuscript is now being handed over to our production team.

Kind regards,

on behalf of

Dr. Andrew Curtis

Academic Editor

PLOS ONE